# Bone Marrow Origin of Mammary Phagocytic Intraductal Macrophages (Foam Cells)

**DOI:** 10.3390/ijms26041699

**Published:** 2025-02-17

**Authors:** Sanford H. Barsky, Krista Mcphail, Justin Wang, Robert M. Hoffman, Yin Ye

**Affiliations:** 1Department of Pathology, Anatomy and Cell Biology, The Clinical and Translational Research Center of Excellence, Meharry Medical College, 1005 Dr. D.B. Todd Jr. Boulevard, Nashville, TN 37208, USA; yye@mmc.edu; 2Star Diagnostics Laboratories, 215 E Warm Springs Rd., Ste 108, Las Vegas, NV 89119, USA; kristamariemcphail@gmail.com; 3Scripps Mercy Hospital, MER 35, San Diego, CA 92103, USA; 4AntiCancer, Inc., 7917 Ostow St., Suite B, San Diego, CA 92111, USA; all@anticancer.org; 5Department of Surgery, University of California at San Diego, 9300 Campus Point Drive, #7220, La Jolla, CA 92037, USA

**Keywords:** mammary intraductal macrophages, foam cells, transgenic mice, bone marrow transplantation, chromogenic markers, microsatellite polymorphisms, X,Y chromosome FISH

## Abstract

Mammary intraductal macrophages (foam cells) in humans are the most commonly encountered cells in spontaneous breast nipple discharge, nipple aspirate fluid, and ductal lavage, yet their origin remains unproven. These cells, in both humans and murine model systems, increase in pregnancy, pseudopregnancy, and other conditions like proliferative fibrocystic disease and intraductal neoplasia, ductal carcinoma in situ (DCIS), where there is intraductal ectasia and obstruction. Previous immunocytochemical studies with macrophage (CD68, lysozyme), epithelial (cytokeratin, estrogen receptor), and myoepithelial (smooth muscle actin, CALLA, maspin) markers have indicated that intraductal foam cells are of macrophage lineage. These foam cells engage in phagocytosis of both endogenous and exogenous substances present within the ducts and are not proliferative. Although it has been suggested that foam cells could derive from tissue-specific and niche-specific precursors or circulating monocytes, to date no experimental nor clinical studies have provided direct proof of their origin. In this study, we provide evidence in both human and murine bone marrow transplant studies that intraductal foam cells are bone marrow-derived. We first studied a registry of sex-mismatched bone marrow transplant recipients who later in life had undergone breast biopsies for either proliferative fibrocystic disease, DCIS, or gynecomastia, and studied these biopsies by XY chromosome fluorescence in situ hybridization (FISH) and informative microsatellite polymorphic markers. The intraductal foam cells were of bone marrow donor-origin. Then, in the experimental bone marrow transplant murine studies, donor marrow from female ROSA26 containing the lacZ reporter were transplanted into either irradiated female recipient transgenic mice carrying the highly penetrant *MMTV-pymT* or FVB/N background mice, where induced pluripotent stem (iPS) cells derived from tail vein fibroblasts of FVB/N-Tg(MMTV-*PyVT*)634Mul/J mice were subsequently injected into their mammary fat pads. In all of the transplanted recipient mice, the intraductal foam cells expressed the β-galactosidase (lacZ) reporter and also co-expressed markers of myeloid–macrophage lineage. The number of donor-derived intraductal foam cells increased in pseudopregnancy 5-fold and in intraductal neoplasia 10-fold. Although macrophages of different origins and lineages are undoubtedly present within both the murine and human breasts, those macrophages that qualify as phagocytic intraductal foam cells are bone marrow-derived.

## 1. Introduction

Mammary intraductal “foam cells” in humans are the most commonly encountered cells in spontaneous breast nipple discharge, nipple aspirate fluid, and ductal lavage, yet their origin remains unproven [1,2,3]. These cells, in both humans and murine model systems, increase in pregnancy, pseudopregnancy, and other conditions like proliferative fibrocystic disease and intraductal neoplasia, ductal carcinoma in situ (DCIS) where there is intraductal ectasia and obstruction. Previous immunocytochemical studies with macrophage (CD68, lysozyme), epithelial (cytokeratin, estrogen receptor), and myoepithelial (smooth muscle actin, CALLA, maspin) markers have indicated that intraductal foam cells are of macrophage lineage [4,5,6]. Mammary macrophages of different classes, different lineages, and different proliferative capacities exist within both murine and human mammary glands [7,8,9,10,11,12,13]. Recent studies, especially in mice, have confirmed the existence of these different macrophage subsets, one derived from embryonic tissue-specific macrophage precursors and one derived from circulating monocytes. Each of these macrophage subsets may have different functions and act within different niches. These studies have largely been based on single-cell transcriptome and biomarker comparisons [11,12,13]. It is thought, based on some of these recent studies, that mammary macrophages recruited in response to lactation and/or neoplasia are derived from circulating monocytes [7,8,9,10,11,12,13]. If one focuses, however, solely on mammary intraductal macrophages in humans and in mice in both the presence and absence of lactation or neoplasia, there has been no experimental nor clinical study that has directly studied their origin. For these reasons we conducted the present study.

## 2. Results

### 2.1. Human Studies

#### 2.1.1. Properties of Human Intraductal Foam Cells

Mammary intraductal “foam cells” in humans often outnumbered exfoliated epithelial cells even in situations where epithelial cells proliferated (Figure 1A). Both the number and density of intraductal foam cells increased both in conditions of ductal ectasia or conditions of intraductal proliferation, or both (Figure 1B,C). In our present study, intraductal foam cells expressed solely macrophage lineage-specific markers (CD68^+^) (Figure 1B). In our study, intraductal foam cells were also non-proliferative (negative Ki-67 immunoreactivity) even in situations where they were juxtaposed to proliferating intraductal epithelial cells (Figure 1C). To strengthen the significance of these findings, we investigated, in addition to Ki-67, PCNA and MCM-2, two other markers expressed at slightly different levels at varying stages of the cell cycle, but never during G_0_, and observed that intraductal foam cells were completely negative for PCNA and MCM-2 as well, suggesting that they are, in fact, in G_0_ and not proliferative (Figure 1C). Although intraductal foam cells phagocytize naturally occurring endogenous lactating and non-lactating secretions, accounting for their characteristic “foamy” appearance, they also phagocytize exogenous substances. In studies on mastectomy specimens following breast-duct endoscopy and installation of barium, gastrograffin, and methylene blue through nipple orifices, foam cells phagocytized these exogenous substances within 1 h (Figure 1D).

#### 2.1.2. Frequency and Density of Human Intraductal Foam Cells

Using SRAs, which initially recognized ductal profiles based on circumferential myoepithelial marker (maspin) immunoreactivity (Figure 2A), processed and expanded these ductal images based on epithelial marker (E-cadherin) immunoreactivity, and then detected intraluminal macrophage (CD68) immunoreactivity (Figure 2B), the frequency and density of intraductal foam cells was calculated. Foam cells increased in conditions of both duct ectasia (which included both non-proliferative and proliferative fibrocystic disease) approximately 5-fold (*p* < 0.01) and intraductal carcinoma approximately 10-fold (*p* < 0.1) (Figure 2C).

#### 2.1.3. Origin of Human Intraductal Foam Cells

In the five cases where breast biopsies were available in patients who had received a sex-mismatched bone marrow transplant years earlier, X,Y chromosomal FISH studies revealed that the intraductal foam cells were of donor origin (Figure 3A–D).

Microsatellite analysis on these same cases was carried out by genomic DNA extraction, PCR, and capillary gel electrophoresis. A panel of informative micro-satellite markers was screened (Table 1) and several were selected. 

Inspection of the electropherograms revealed allelic variability, which distinguished donor from recipient and confirmed the bone marrow origin of the intraductal foam cells (Figure 4A,B).

In the human transplant studies, we used two well proven methods to distinguish donor from recipient cells, X,Y chromosome FISH, and informative DNA microsatellite polymorphisms. With both methodologies, the intraductal foam cells appeared to be of donor origin. Each methodology has inherent limitations. For example, in X,Y chromosome FISH, there can be spontaneous loss of the Y chromosome in proliferating cells. In DNA microsatellite polymorphism analysis, although the marker may be informative, there may be spontaneous loss of heterozygosity in the area exploited, which may confound the results. Although both limitations apply to situations where the cells are proliferating and especially to neoplasia, these limitations are less likely here where the intraductal macrophages are thought to be terminally differentiated and non-proliferating. Also, spontaneous homotypic and heterotypic cell fusion can confound the results, but there is no evidence to believe that this phenomenon has occurred in the setting of intraductal macrophages. The fact that our observations using two different methodologies both confirmed the donor origin of the intraductal macrophages strengthens our conclusions.

### 2.2. Murine Studies

#### 2.2.1. Properties of Murine Intraductal Foam Cells

Murine intraductal foam cells resembled their human counterparts in terms of appearance, expression of macrophage markers, and increased density in conditions of ductal ectasia and ductal carcinoma (Figure 5A–C).

#### 2.2.2. Frequency and Density of Murine Intraductal Foam Cells

Using algorithmic image analysis similar to the methods used in the human studies, murine intraductal foam cells increased in ductal ectasia from pseudopregnancy approximately 5-fold (*p* < 0.1) and transgene-induced intraductal carcinoma approximately 10-fold (*p* < 0.1) (Figure 5D).

#### 2.2.3. Origin of Murine Intraductal Foam Cells

Both FVB/N-Tg(MMTV-*PyVT*)634Mul/J and FVB/N mice whose cleared mammary fat pads were injected with iPS cells containing MMTV-*PyVT*, all of whom were recipients of a tagged donor bone marrow transplant (Figure 6A–C), exhibited intraductal foam cells of donor origin (Figure 7A–D). We then strengthened our conclusions with regard to the origin and nature of the murine intraductal foam cells by comparing the number of CD68^+^ intraductal foam cells with and without BM markers. We conducted dual labeling of the intraductal foam cells that expressed the β-galactosidase (LacZ)^+^ tag with an alkaline phosphatase-conjugated secondary goat antibody to both rabbit anti-mouse CD68 and rabbit anti-mouse *CD11b*, both detected by 1-Step™ NBT/BCIP Substrate Solution, giving a red color. Dual blue/red signals with either antibody were observed within the same intraductal foam cells, suggesting that in these murine studies, virtually 100% of the intraductal foam cells were of both bone marrow donor-origin and myeloid–macrophage lineage (Figure 7D). No intraductal foam cells were of local origin. Although we did note CD68^+^ cells that were β-galactosidase (LacZ)^−^ and CD68^-^ cells that were β-galactosidase (LacZ)^+^, they were located within the mammary stroma and not within the intraductal lumens. Undoubtedly, they represented, respectively, resident macrophages of local origin and non-macrophages, e.g., neutrophils and lymphocytes of bone marrow origin. We did of course observe β-galactosidase (LacZ)^+^ cells within mammary stroma (Figure 7B), which could represent bone marrow-derived macrophages in transit to become intraductal foam cells.

## 3. Discussion

Mammary intraductal macrophages (foam cells) in humans are the most commonly encountered cells in spontaneous nipple discharge, nipple aspirate fluid, and ductal lavage, yet their precise origin and significance remain a mystery [1,2,3,4,5,6,7,8,9,10,11,12,13]. These cells are thought to increase in both numbers and density in human pregnancy and other conditions of ductal ectasia and/or obstruction [1,2,3,4,5,6,7,8,9,10]. Our findings in this study confirm these observations. Although it has been suggested that intraductal foam cells in humans could originate from either tissue-specific embryonic or bone marrow-derived precursors, to date, no clinical study has provided direct proof of their origin [7,8,9,10,11,12,13]. To our knowledge, ours is the first study that provides direct proof in studies obtained from a human bone marrow transplant registry that human intraductal foam cells are bone marrow-derived.

Although there have been recent murine studies that have used flow cytometry, single cell RNA seq, advanced three-dimensional intravital imaging and transgenic reporter mice crosses that have supported the conclusion that lactation-induced macrophages (liMacs) are derived from circulating monocytes [7,8,9], ours is the first study that uses tagged donor marrow murine transplantation to show that murine intraductal foam cells are also indeed bone marrow-derived.

In both our human and murine studies [14,15,16,17,18,19,20,21,22,23,24,25,26,27,28,29,30,31], we examined situations where the bone marrow transplant occurred well before the onset of puberty, pseudopregnancy, mammary morphogenesis and oncogenesis. Clearly during mammary gland development, accompanying the development of the ductal–lobular system in all these situations, the pool of mammary macrophages would be expected to expand and we wanted to observe the ultimate source of this expansion, especially with respect to intraductal foam cells.

Intraductal foam cells appear to be non-proliferative in our study by three different and independent markers of the cell cycle: Ki-67, PCNA, and MCM-2, markers expressed at slightly different levels at varying stages of the cell cycle but never during G_0_ [32,33,34,35,36,37]. So, the apparent increase in the number of intraductal foam cells in conditions of ductal ectasia or intraductal proliferation (fibrocystic disease or DCIS) in humans suggests that they migrate intraductally from either interductal, periductal, or stromal locations. Macrophages in these source locations are not foamy in nature; hence, they only acquire their foamy appearance from their phagocytosis of endogenous materials secreted by the ducts when the macrophages become intraductal foam cells. Our demonstration of active phagocytosis of exogenous materials, e.g., barium, instilled into ducts through nipple orifices prior to mastectomy in humans, also supports this conclusion of active phagocytosis.

Pregnancy also causes major changes to the physiology of the mammary gland. Within days of conception, the epithelium tree undergoes a dramatic proliferative phase that continues through pregnancy [38,39,40]. At that time, the extended epithelium network is fully functional to produce milk. Therefore, it would be anticipated that mammary intraductal foam cells would increase in pregnancy and pseudopregnancy since there are more intraductal secretions to phagocytize, and this is what we observed also in our murine studies.

Intraductal foam cells also increase in human DCIS [10] as well as in murine ducts of FVB/N-Tg(MMTV-*PyVT*)634Mul/J transgenics containing evolving mammary carcinoma [8]. In both situations it would be expected that ductal obstructions from carcinoma would cause accumulations of ductal secretions and increased phagocytosis.

Much more is known about murine mammary macrophages, in general, and in murine intraductal foam cells in particular, than is known about their human counterparts. This is largely due to the aforementioned highly sophisticated murine studies, which have more precisely defined macrophage subsets in exquisite detail [7,8,9]. Mammary gland macrophages are heterogeneous in origin and phenotype and carry out many organ-specific functions [7]. Many tissue-resident macrophage populations are thought to be derived from embryonic precursors and self-renew locally [11], while others are continuously replenished by circulating monocytes. Macrophage residency, density, and proliferation can be partially independent or dependent on bone marrow contribution but their dependency on bone marrow contribution is reflective of their location within the mammary gland and specifically their function. In virgin mammary glands, stromal macrophage populations predominate, but in the lactating mammary gland, ductal macrophages dominate and form a continuous layer between myoepithelial and epithelial luminal cells [8]. These ductal macrophages differ from stromal macrophages by flow cytometric and single-cell RNA-seq studies [9] and are likely derived from circulating monocytes.

Murine mammary macrophages specifically associated with lactation (liMacs) have recently been profiled [7]. liMacs were found predominantly within ductal lumens or in juxtaposition to the luminal surface of the basal layer. Using transgenic reporter mice crosses, which allowed the fate-mapping of monocytes and granulocytes, these liMacs were shown to arise from circulating monocytes and not tissue-resident macrophages [7]. liMacs then migrated into ductal lumens and lactational secretions in contrast to the behavior of typical tissue-resident macrophages, which usually did not migrate. Using additional transgenic reporter mice crosses and single-cell RNA-seq, it was shown that liMacs not only express genes associated with migration, lipid metabolism, and phagocytosis, but can self-proliferate and expand when monocyte egress from the bone marrow was blocked [7]. Therefore, these recent studies also support our findings in both mice and humans that intraductal foam cells are indeed bone marrow-derived, engage in phagocytosis largely of endogenous lipid substances, and migrate into ductal lumens. The only difference in our findings was that we observed that intraductal foam cells lack proliferative activity. However, since the liMacs with proliferative activity comprised only a small subset of overall liMacs [7], the interductal and periductal macrophages before they migrate intraductally could well be proliferative and then lose this ability to proliferate.

Interestingly, by single-cell RNA-seq, liMacs also express genes of immunosuppression. Since the lactating breast is prone to bacteria mastitis, this raises the possibility that liMacs are responsible [7,8,9].

By single-cell RNA-seq, ductal macrophages and liMacs also strongly resemble mammary tumor-associated macrophages (TAMs), which likely also arise from circulating monocytes and not tissue-resident macrophages. Just as it has been noted that intraductal foam cells associated with lactation may be immunosuppressive, intraductal foam cells associated with human DCIS [10] or ducts distended with carcinoma in FVB/N-Tg(MMTV-*PyVT*)634Mul/J transgenics [8] may also be immunosuppressive. These intraductal foam cells, therefore, are TAMS in the truest sense of the word and bone marrow-derived.

Our study focuses on intraductal macrophages (foam cells), which lie within lactating ducts and are juxtaposed to ductal neoplastic processes. This is why we initially raised their putative role in immunosuppression and not wound healing per se. But certainly, the macrophages that end up as intraductal foam cells but begin as circulating bone marrow-derived monocytes first have had to enter the stromal compartment of the breast. At that stage they may, in fact, show a strikingly different phenotype including both proliferative and wound-healing potential responsible for both tissue repair and remodeling.

In the recruitment of mammary TAMs, compelling recent experimental and clinical evidence has demonstrated that breast cancers are thought to alter the transcriptome of circulating human monocytes [13]. Furthermore, TAMs are transcriptionally distinct from normal monocytes and their respective tissue-resident macrophages. Breast cancer perturbates the macrophage landscape and negatively affects the outcome, suggesting that most macrophages associated with cancer are both protumorigenic as well as immunosuppressive macrophages [12,13].

But are intraductal foam cells also protumorigenic and immunosuppressive? Some recent studies have suggested that the transcriptome of lactational intraductal foam cells overlap with the transcriptome of TAMs [7,8,9], suggesting that a subset of intraductal foam cells function as TAMs. Intraductal foam cells juxtaposed to human DCIS or murine intraductal carcinoma would therefore be equally protumorigenic. Recent Phase 1 clinical trials attempting to convert protumorigenic macrophages to anti-tumorigenic macrophages by blocking the CD47-SIRPα axis, for example, and making macrophages less immunosuppressive and more phagocytic of tumor cells themselves, have shown promise [12]. This repolarization of macrophages to an anti-tumorigenic phenotype could hypothetically be applied to intraductal foam cells to make them act as DCIS sentries.

## 4. Materials and Methods

### 4.1. Human Studies

#### 4.1.1. Ethics Approval and Consent to Participate

Access to mastectomy specimens following consented breast-duct endoscopy, approved by the UCLA Human Subjects Protection Committee and the US Food and Drug Administration (IDE G940002), was reported in previous studies [14,15]. In the present study, we examined further formalin fixed, paraffin-embedded (FFPE) materials collected and deidentified from that study.

Collection and use of human breast cancer tissues, completely anonymized, had also been approved by the Ohio State University Cancer Institutional Review Board (IRB) under protocol 2006C0042, date of approval 04/10/06. Additional cases of breast cancer were obtained and anonymized from the Meharry Medical College Translational Pathology Shared Resource Core, IRB Protocol 23-10-1410, date of approval 11/15/24.

Access to the Ohio State and Children’s Hospital Transplant Registry and the obtaining of both donor and recipient tissues and DNA, de-identified following patient consent, was approved by the Ohio State University Cancer IRB under protocol 2006C0042, date of approval 04/10/06.

#### 4.1.2. Selection of Cases

A total of 5 cases from the original breast ductoscopy cohort (UCLA) and 25 cases each of normal breast tissues (reduction mammoplasties), non-proliferative fibrocystic disease containing ductal ectasia, proliferative fibrocystic disease, ductal carcinoma in situ, and/or invasive cancer (100 cases in total) (Ohio State and Meharry Medical College) were selected for study. After searching our transplant database both at the Ohio State University College of Medicine and the neighboring Children’s Hospital that contained records of individuals who had received a sex-mismatched bone marrow transplant for non-Hodgkin’s lymphoma or leukemia, who survived and years later developed breast disease (ductal ectasia, proliferative fibrocystic disease, DCIS or invasive cancer or gynecomastia), we found a total of 5 cases.

#### 4.1.3. Histological, Immunofluorescence and Immunohistochemistry Studies

Primary antibodies used in human studies included antibodies to Ki-67 (Spring Bioscience, Pleasanton, CA, USA), PCNA (Novus Biologicals, Centennial, CO, USA), MCM-2 (LSBio, Seattle, WA, USA), CD68 (Cell Signaling Technology, Inc., Danvers, MA, USA) and maspin (Novus Biologicals), all rabbit anti-human. The appropriate secondary antibodies were goat anti-rabbit, linked to HRP and the chromogen, DAB (Rockland Immunocytochemicals, Limerick, PA, USA). Other primary antibodies used included Alexa Fluor 488 conjugated mouse anti-human E-cadherin 24E10 (Cell Signaling Technology, Inc.) and rabbit antibody to human maspin (Novus Biologicals) followed by Alexa Fluor 594-conjugated goat anti-rabbit (Thermo Fisher Scientific, Inc., Waltham, MA, USA) followed by DAPI, Vector Laboratories, Newark, CA, USA).

#### 4.1.4. Tissue Microarray (TMA) Construction and Image Algorithmic Analysis

Multiple 2 mm tissue cores of tumor from each paraffin-embedded donor block (average of 10 cores/block) were arrayed into recipient TMA blocks. Our specific TMA algorithms carried out virtual alignment, image processing, and the application of the epithelial recognition algorithms (ERAs) and specific recognition algorithms (SRAs), which recognized ductal profiles surrounded by myoepithelial cells (maspin cytoplasmic immunoreactivity and immunofluorescence), adjacent luminal epithelial cells (E-cadherin membrane immunofluorescence), DAPI nuclear fluorescence, and intraluminal foam cells (CD68^+^ cytoplasmic immunoreactivity), based upon cytoplasmic/membrane compartmental immunofluorescence and immunoreactivities. Their respective signal intensities [16,17,18,19] were quantitated utilizing ImageJ [20]. Image acquisition was acquired by the iSCAN System (BioImagene, Inc., Cupertino, CA, USA).

#### 4.1.5. X,Y Chromosomal FISH Studies

FISH for the presence or absence of an X or Y chromosome (CEP X SpectrumGreen/CEP Y SpectrumOrange) was performed using the AneuVysion (Vysis CEP X, Y-alpha satellite Multicolor Probe Panel, (Abbott, Abbott Park, IL, USA)) to detect alpha satellite sequences in the centromere regions of the chromosomes in accordance with the manufacturer’s guidelines [21,22] and performed manually. Briefly, formalin fixed, paraffin-embedded tissues were cut into 3 to 4 µm-thick sections, incubated over night at 56 °C, deparaffinized, washed, digested with protease, formalin fixed, and denatured. The slides were then hybridized with the manufacturer’s recommended probe at 37 °C for 16 h. The slides were then washed in a post-hybridization wash, counter stained with 4′-6-Diamidino-2-phenylindole (DAPI), and covered with a coverslip. Specimens were evaluated with the Olympus BX51 microscope (Olympus Optical Company, Ltd., Hachioji, Tokyo, Japan) under oil immersion at ×100 magnification using the recommended filters. The presence or absence of an X or Y chromosome signal in at least 60 interphase nuclei with non-overlapping nuclei in the intraductal foam cells and adjacent tissues was determined.

#### 4.1.6. Laser Capture Microdissection Studies

Paraffin embedded sections (8 µm) of human breast biopsies were obtained, fixed in 70% ethanol, stained with hematoxylin, and progressively dehydrated. Intraductal foam cells were microdissected using a Pixcell II Laser Capture Microdissection 788 Laboratory System (Arcturus, Inc., Mountain View, CA, USA) [16] and stored at −80 °C. At least 10–100 intraductal foam cell clusters were obtained from each case and processed.

#### 4.1.7. Microsatellite Polymorphism Studies

To further confirm the donor bone marrow origin of the intraductal macrophages in the aforementioned cases, we conducted DNA fingerprinting studies [23,24]. We initially carried out laser capture microdissection of the respective areas within the paraffin blocks to ensure that the given histological sampling was reasonably “pure”. DNA fingerprinting with a series of highly informative microsatellite markers was then used to “DNA fingerprint” the initial donor (bone marrow), the laser-captured intraductal foam cells, and the adjacent breast tissues (host). In these studies, we examined a number of different polymorphic loci, which both had an inherent low rate of loss of heterozygosity (%LOH) but also a high informative rate (%Inf) (Table 1).

Genomic DNA was extracted from both donor and recipient tissue sources. PCR amplification and capillary gel electrophoresis was subsequently carried out. A panel of several microsatellite markers was used for analysis. PCR amplification was performed using a Bio-Rad Thermal Cycler (Bio-Rad Laboratories, Hercules, CA, USA) in 10 µL reaction mixtures containing 20 ng of genomic DNA, 0.4 µM of each primer, and 1× MyTaq Mix according to the recommendations of the manufacturer. Labeled (with FAM or HEX) 5′-fluorophore of each forward primer was used. The products were analyzed on an Applied Biosystems™ GeneScan™ (Thermofisher, Waltham, MA, USA). Inspection of the electropherograms revealed allelic variability, which distinguished donor from recipient.

### 4.2. Murine Studies

#### 4.2.1. Ethics Approval

All animals used in the study were purchased by grants and owned by the university. Therefore, the animals used were not privately owned. All animal studies including the bone marrow transplant studies were approved by the Ohio State University’s Animal Care and Use Committee (IACUC), protocols 2004A0179 and 2007A0218 (dates of approval 10/15/04 and 02/20/07) and by the Ohio State University’s Institutional Biosafety Committee, protocol 2007R0057, (date of approval 02/20/07). The mice were treated in compliance with the NIH and internal IACUC guidelines. Continued animal studies were approved by the University of Nevada’s School of Medicine and the Nevada Cancer Institute’s IACUC, protocols 00439 and 00440 (date of approval 09/18/09). Final animal studies were conducted under an Interinstitutional Agreement between the California University of Science and Medicine and Anticancer, Inc. using the latter’s IACUC protocol D16-00503 and OLAW A3873-01 (date of approval 03/28/20).

#### 4.2.2. Initial Murine Studies

A total of 40 each of 4-week-old female FVB/N-Tg(MMTV-*PyVT*)634Mul/J and FVB/N mice were obtained (The Jackson Laboratory Biomedical Research Institute, Bar Harbor, ME, USA). Previously iPS cells containing MMTV-*PyVT* were created from tail vein fibroblasts of the FVB/N-Tg(MMTV-*PyVT*)634Mul/J and found, when injected into a cleared mammary fat pad of FVB/N mice, to form a mammary gland containing both intraductal as well as invasive ductal carcinoma [25,26]. Ten ROSA26 mice, which constitutively express β-galactosidase (LacZ) served as bone marrow donors [27].

#### 4.2.3. Bone Marrow Transplantation Studies

Bone marrow of ROSA 26 donor mice was harvested by femoral flushing, placed in culture to remove the non-adherent cells, and transplanted into lethally irradiated recipient mice (irradiated with 10 Gy at 4–8 weeks of age). Recipient mice included both the FVB/N-Tg(MMTV-*PyVT*)634Mul/J (10 mice) and FVB/N (20 mice). Recipient mice received 1–5 × 10^6^ genetically marked cells through the tail vein according to standard protocols [28,29]. Transplanted mice were monitored over the next 2–3 weeks and evidence of engraftment was determined by examining peripheral blood or bone marrow for colorimetric detection of the β-galactosidase (LacZ) marker. Mice designated with successful bone marrow engraftment showed at least 50% donor marrow engraftment after 60 days.

#### 4.2.4. Subsequent Murine Studies

Select recipient FVB/N (10 mice) were then made pseudo-pregnant with 60-day release pellets of 2 mg 17β-estradiol and 2 mg 17α-hydroxypregnenolone (Innovative Research of America, Sarasota, FL, USA).

In the two previously described sets of experiments involving the transfer of tagged bone marrow, one set was transplanted into 10 FVB/N-Tg(MMTV-*PyVT*)634Mul/J transgenic mice, which spontaneously developed mammary carcinomas at 60–90 days, and a second set transplanted into 10 FVB/N mice whose cleared mammary fat pads [30] were then subsequently injected with iPS cells derived from tail vein fibroblasts of FVB/N-Tg(MMTV-*PyVT*)634Mul/J mice. In the latter set, we did not inject the iPS cells until 6 weeks after the bone marrow transplant. In both sets of experiments, we harvested the tumors at two time points, when the tumors initially emerged and after they had reached 2 cm in size. Both time points were well beyond the six week time period of successful reconstituted bone marrow transfer. Selected mice were then euthanized with CO_2_ inhalational anesthesia.

In this manner we attempted to optimize our detection of any marrow transplant-derived cells that may have been recruited to the breast either in the settings of pseudopregnancy or emerging breast cancer.

#### 4.2.5. Necropsy and TMA Studies

Extirpated murine tissues were formalin-fixed and paraffin-embedded, made into TMAs, and analyzed with the aforementioned imaging algorithms designed to detect and quantitate intraductal foam cells in a similar manner as that used in human studies. Selected tissues were also subjected to X-gal staining [31].

#### 4.2.6. X-Gal Staining

Selected tissues were fixed for 1 h in 4% formaldehyde in PBS, washed three times in rinse buffer (2 mM MgCl_2_/0.1% sodium deoxycholate/0.2% Nonidet P-40 in PBS) and rotated in X-Gal staining solution (1 mg/mL X-Gal, 5 mM potassium ferricyanide and 5 mM potassium ferrocyanide in rinse buffer) at 37 °C for 18 h, washed in PBS, and processed for whole mounting [31]. This staining procedure was able to unequivocally distinguish donor cells (which turned blue) from recipient cells, which displayed only counterstain.

#### 4.2.7. Histological and Immunohistochemistry Studies

Primary antibodies used in the murine studies included antibodies to Ki-67 (Spring Bioscience, Pleasanton, CA, USA), PCNA (Novus Biologicals, Centennial, CO, USA), MCM-2 (LSBio, Seattle, WA, USA), CD68 (Cell Signaling Technology, Inc., Danvers, MA, USA), and CD11b (Novus Biologicals), all rabbit anti-murine. The appropriate secondary antibodies were goat anti-rabbit, linked to HRP and the chromogen, DAB and goat anti-rabbit, linked to alkaline phosphatase, and the chromogen, 1-Step™ NBT/BCIP substrate solution (Rockland Immunochemicals, Inc., Limerick, PA, USA). Dual labeling with either anti-CD68 or anti-CD11b followed by goat anti-rabbit, linked to alkaline phosphatase, was then carried out on the selected tissues previously subjected to X-gal staining.

#### 4.2.8. General Statistical Analysis Studies

Illustrated photomicrographs depicting histopathology, immunocytochemistry, fluorescence and colorimetry in either the human and/or murine studies were representative of our typical results. A total of 100 ductal profiles/biopsy from both the human cases as well as the murine cases were studied. All in vivo (murine) experiments were performed in groups of 10 mice. All individual in vitro experiments were replicated five times. Within each experiment, five technical replicates were also conducted. Representative results depicted means ± standard deviations. All stated or calculated differences implied differences in statistical significance, assessed by the two tailed students *t* test as well as ANOVA.

## 5. Conclusions

As has been stated, our study focuses on the intraductal foam cell and the evidence supporting its exclusive myeloid–macrophage lineage and bone marrow origin. Since the phagocytic appearance of the foam cell is not apparent until it enters its intraductal location, other phenotypic properties of this cell, e.g., immunosuppression, may only be expressed when it reaches its intraductal location. Alternatively, the subset of bone marrow-derived myeloid–macrophage lineage cells destined to become intraductal foam cells, which also participate as TAMs in the setting of DCIS may have acquired its phenotypic properties at an earlier stage ranging from its bone marrow origin, circulating monocyte pool, or entrance into the mammary stroma. All these possibilities merit investigation in future studies.

## Figures and Tables

**Figure 1 ijms-26-01699-f001:**
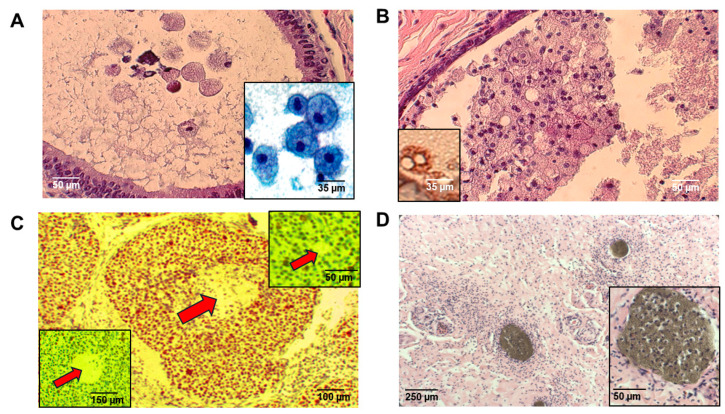
Features of human mammary intraductal foam cells. Mammary intraductal “foam cells” are depicted histologically in a tissue section (**A**) and cytologically within nipple aspirate fluid (**A**, right lower inset). In ductal ectasia, their number substantially increased (**B**) but they consistently exhibited their CD68^+^ macrophage lineage (**B**, left lower inset). Intraductal foam cells in the center of a duct (arrow) with adjacent proliferating DCIS exhibited completely negative Ki-67 immunoreactivity (**C**). Intraductal foam cells were also completely negative for both PCNA (arrow) (**C**, left lower inset) as well as MCM-2 (arrow) (**C**, right upper inset), suggesting that they are, in fact, in G_0_. Intraductal instillation of barium (**D**) was followed by phagocytosis by intraductal macrophages (**D**, right lower inset). Scale bars are provided.

**Figure 2 ijms-26-01699-f002:**
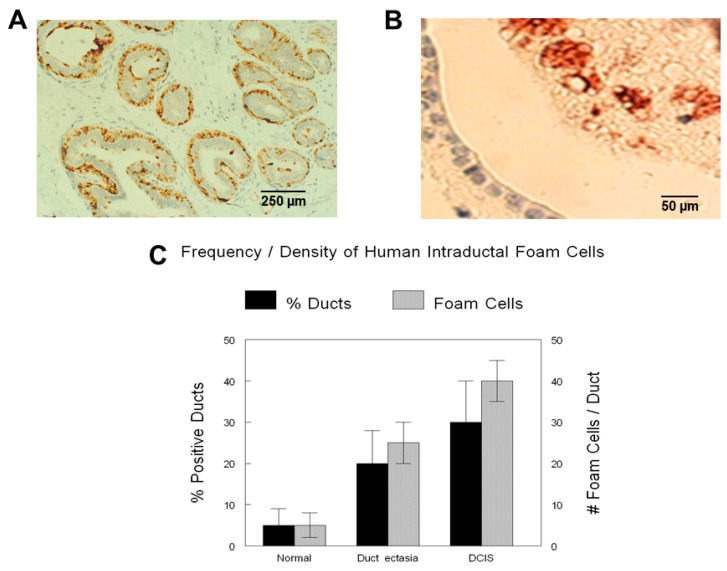
Imaging algorithmic determinations of human foam-cell density. Our specific TMA algorithms, which carried out virtual alignment, image processing, and the application of the epithelial recognition algorithms (ERAs) and specific recognition algorithms (SRAs), which recognized ductal profiles based initially on circumferential maspin myoepithelial immunoreactivity (**A**), and recognition of intraductal macrophage CD68 immunoreactivity (**B**) were able to calculate frequency and density of intraductal foam cells (**C**). % ducts (% positive ducts) are the overall percentage of ducts that contain intraductal macrophages (foam cells). Foam cells (# of foam cells/duct) are the average numbers of intraductal macrophages (foam cells)/duct in the ducts containing foam cells. Scale bars are provided.

**Figure 3 ijms-26-01699-f003:**
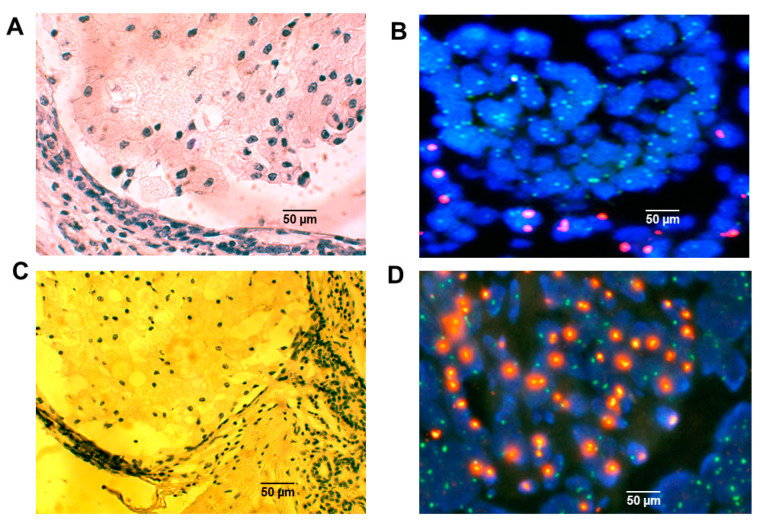
X,Y chromosomal FISH studies on human transplant cases. FISH for the presence or absence of an X or Y chromosome (CEP X SpectrumGreen/CEP Y SpectrumOrange) in sections of intraductal foam cells from a male who had received a female bone marrow donor (**A**,**B**) and a female who had received a male bone marrow donor (**C**,**D**) identified the Y chromosome as fluorescing bright orange and the X chromosome as fluorescing green (**B**,**D**). In each case the intraductal foam cells were of bone marrow donor-origin. Scale bars are provided.

**Figure 4 ijms-26-01699-f004:**
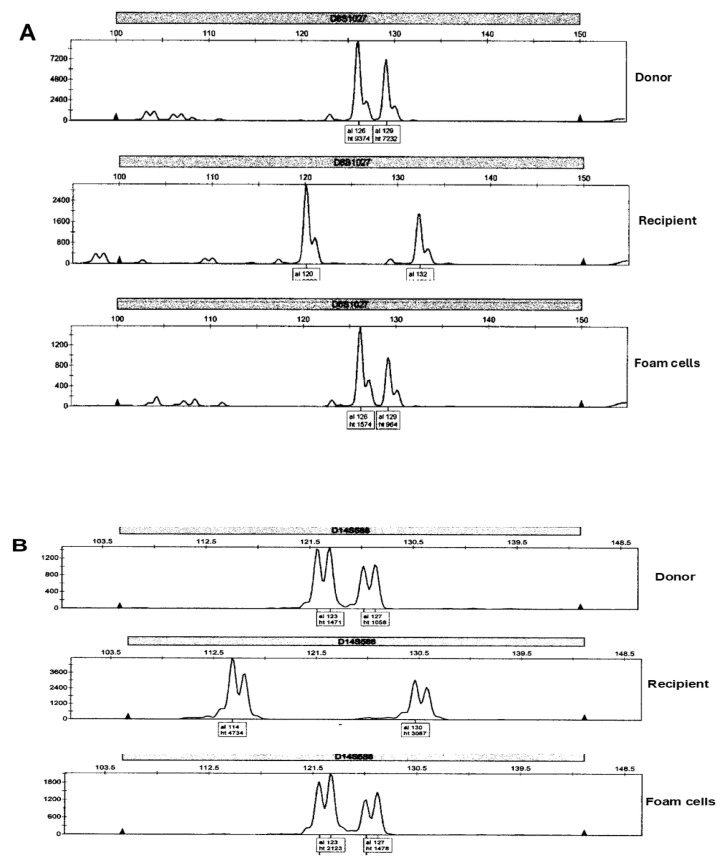
Microsatellite donor/recipient heterozygosity. GeneScan electropherograms of microsatellite PCR (D6S1027) (**A**) and D14S588 (**B**), respectively, of two individual cases distinguish donor from recipient and confirm the donor bone marrow origin of the intraductal foam cells. The abscissas of each panel show the allele sizes in bp and the ordinates show the allele peak heights in arbitrary fluorescence units.

**Figure 5 ijms-26-01699-f005:**
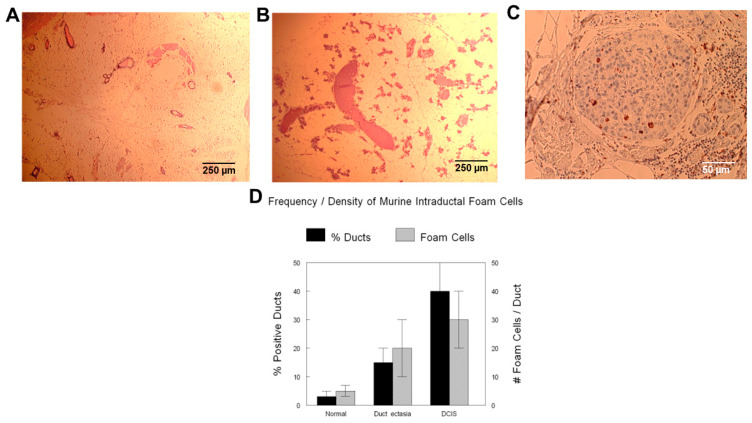
Features of murine mammary intraductal foam cells. The prepubertal murine mammary gland is relatively devoid of ducts (**A**), which both proliferate and dilate in puberty, pregnancy, and conditons of pseudopregancy (**B**). Mammary intraductal foam cells identified by CD68 immunoreactivity increase during the latter and in conditons of intraductal carcinoma (**C**). Imaging processing algorithms similar to those used in the human studies were able to calculate frequency and density of intraductal foam cells (**D**). % ducts (% positive ducts) are the overall percentage of ducts that contain intraductal macrophages (foam cells). Foam cells (# of foam cells/duct) are the average numbers of intraductal macrophages (foam cells)/duct in the ducts containing foam cells. Scale bars are provided.

**Figure 6 ijms-26-01699-f006:**
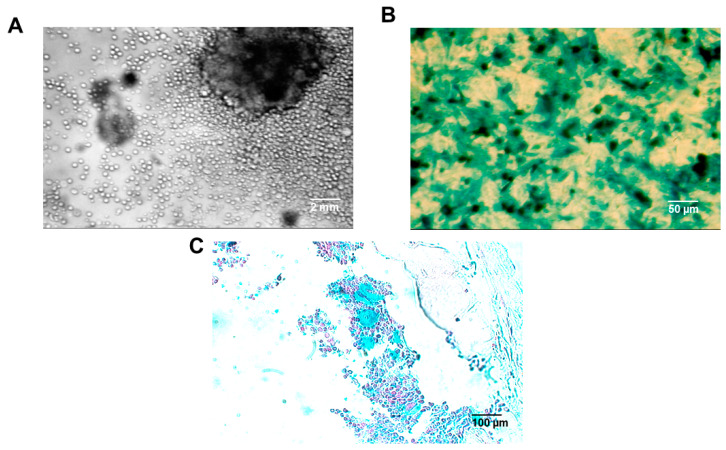
Tagging donor bone marrow with LacZ. Bone marrow of ROSA 26 donor mice, harvested by femoral flushing (**A**) and marked with the β-galactosidase (LacZ) reporter (**B**), was transplanted into recipient mice and successful engraftment was marked by at least 50% engraftment when sections of murine bone marrow were analyzed 60 days after bone marrow transplant (**C**). Gross photographs and photomicrographs are depicted from single representative cases, but 8/10 mice showed successful bone marrow engraftment. Scale bars are provided.

**Figure 7 ijms-26-01699-f007:**
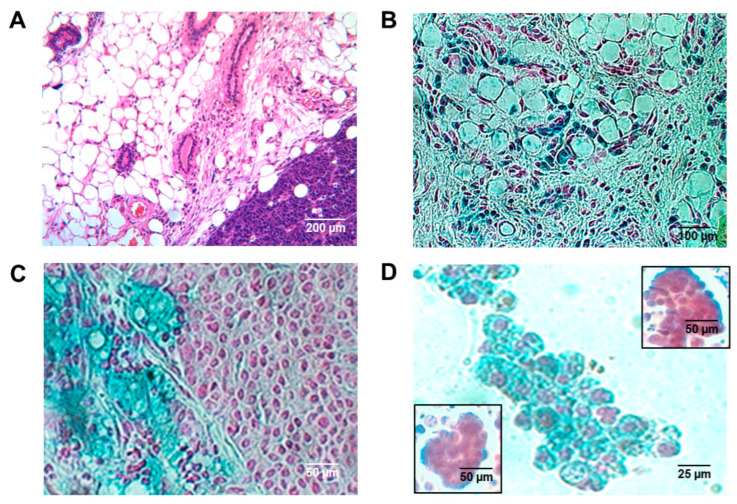
Tagged intraductal foam cells in recipient mice. Recipient FVB/N-Tg(MMTV-*PyVT*)634Mul/J and FVB/N injected with iPS cells containing MMTV-*PyVT* exhibited intraductal carcinoma (**A**) and both stromal macrophages (**B**) and intraductal foam cells (**C**) tagged with the β-galactosidase (LacZ) reporter. Mice made pseudo-pregnant also expressed tagged foam cells in derived nipple fluid (**D**). Dual labeling of these tagged intraductal foam cells with an alkaline phosphatase-conjugated secondary goat antibody to rabbit anti-mouse CD68 (**D**, left lower inset) and rabbit anti-mouse CD11b (**D**, right upper inset) revealed dual blue/red signals within all of the intraductal foam cells, suggesting that virtually 100% of the intraductal cells were of both macrophage–monocyte origin and bone marrow donor-origin and none were of local origin. Scale bars are provided.

**Table 1 ijms-26-01699-t001:** Informative microsatellite loci used to distinguish donor from recipient.

Chr	Locus	% lnf	%LOH	Marker	Allele Size Range	Dye	Transplant Study
1	D1S1596	0.86	0.39	GATA26G09	101–125	FAM	yes
2	D2S1394	0.77	0.46	GATA69E12	144–184	FAM	yes
3	D3S2432	0.78	0.58	GATA27C08	118–170	FAM	yes
5	D5S2500	0.8	0.59	GATA67D03	149–181	HEX	yes
6	D6S1027	0.74	0.52	ATA22G07	110–150	FAM	yes
7	D7S3070	0.89	0.56	GATA189C06	184–208	FAM	yes
8	D8S1132	0.67	0.57	GATA26E03	139–171	FAM	yes
9	D9S1825	0.67	0.64	AFMb029xg1	127–145	FAM	yes
10	D10S1230	0.78	0.50	ATA29C03	113–140	FAM	yes
11	D11S1984	0.80	0.62	GGAA17G05	166–206	HEX	yes
12	D12S1042	0.87	0.53	ATA27A06	118–136	HEX	yes
13	D13S317	0.85	0.61	GATA7G10	175–199	HEX	yes
14	D14S588	0.78	0.50	GGAA4A12	110–145	FAM	yes
15	D15S818	0.77	0.56	GATA85D02	150–170	HEX	yes
16	D16S764	0.94	0.57	GATA42E11	86–126	HEX	yes
17	D17S2193	0.84	0.66	ATA43A10Z	88–123	HEX	yes
19	D19S591	0.83	0.41	GATA44F10	90–120	FAM	yes
20	D20S851	0.69	0.54	AFMa218yb5	128–150	FAM	yes
22	D22S1169	0.71	0.56	AFMb337zh9	118–134	FAM	yes

## Data Availability

All cell lines, tissues, and all data sets generated and used in the study are available upon request. All imaging algorithms are also available.

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
