# Peer review of "Bone Marrow Origin of Mammary Phagocytic Intraductal Macrophages (Foam Cells)"

_ijms, 2025, doi:10.3390/ijms26041699_

Round 1

Reviewer 1 Report

Comments and Suggestions for Authors

In this manuscript, Barsky et al., analyzed the intraductal foam cells from both humans and mice with bone marrow transplantation and confirmed that these intraductal foam cells are bone marrow-derived. This is an interesting topic, although previous studies have suggested the bone marrow origin of these foam cells, this study provides direct evidence that the intraductal foam cells are from bone marrow. The data are clearly presented and methods to address experimental questions are appropriate. Several minor issues are enumerated below.

1.       The explanation in Section 3.2.3 is overly simplistic. A more detailed description of the results, along with an explanation of how the conclusion was reached, is necessary.

 2.       In the mice bone marrow transplantation study, detailed time points should be provided. Specify how many weeks after the bone marrow transfer the iPS cells were injected and when the tissues were collected. Typically, hematopoietic cells are considered fully reconstituted six weeks after bone marrow transfer.

Author Response

  1. The explanation in Section 3.2.3 is overly simplistic. A more detailed description of the results, along with an explanation of how the conclusion was reached, is necessary.

In the human transplant studies, we used two well proven methods to distinguish donor from recipient cells, X,Y chromosome FISH and informative DNA microsatellite polymorphisms. With both methodologies, the intraductal foam cells appeared to be of donor origin. Each methodology has inherent limitations. For example in X,Y chromosome FISH, there can be spontaneous loss of the Y chromosome in proliferating cells. In DNA microsatellite polymorphism analysis although the marker may be informative, there may be spontaneous loss of heterozygosity in the chromosomal loci exploited which may confound the results. Although both limitations apply to situations where cells are proliferating and especially to neoplasia, these limitations are less likely here where the intraductal macrophages are thought to be terminally differentiated and non-proliferating. Also spontaneous homotypic and heterotypic cell fusion can confound the results but there is no evidence to believe that this phenomenon is occurring in the setting of intraductal macrophages. The fact that our observations using two different methodologies both confirmed the donor origin of the intraductal macrophages strengthens our conclusions.

We have clarified all these points in the Results section of our revised manuscript.  

  1. In the mice bone marrow transplantation study, detailed time points should be provided. Specify how many weeks after the bone marrow transfer the iPS cells were injected and when the tissues were collected. Typically, hematopoietic cells are considered fully reconstituted six weeks after bone marrow transfer.

We did not make it clear initially the exact experiments which we conducted. In the two previously described sets of experiments involving transfer of tagged bone marrow, one set was transplanted into FVB/N-Tg(MMTV-PyVT)634Mul/J, transgenic mice which spontaneously develop mammary carcinomas at 60-90 days and a second set transplanted into background FVB/N mice whose mammary fat pads were then subsequently injected with iPS cells derived from tail vein fibroblasts of FVB/N-Tg(MMTV-PyVT)634Mul/J mice. In the latter set of experiments, we did not inject the iPS cells until six weeks after the bone marrow transplant. In both sets of experiments we harvested the mammary carcinomas at 2 time points, when the tumors initially emerged and after they had reached 2 cm in size. Both time points were well beyond the six week time period of successful reconstituted bone marrow transfer. Select mice were then euthanized with CO2 inhalational anesthesia. In this manner we attempted to optimize our detection of any marrow transplant-derived cells that may have been recruited to the emerging breast cancer.

We have clarified these points in the Materials and methods section of the revised manuscript.

Reviewer 2 Report

Comments and Suggestions for Authors

Major critiques:

1)     The bone marrow (BM) origin of mammary gland macrophages is convincing; however, the numbers of CD68+ cells with and without BM markers have not been compared. This is a missed opportunity to clarify the extent of contribution of myeloid-macrophage population from the BM and local sources.

2)     Figure 7 shows examples of cells from BM but there is no verification that these cells (or part of them) are from the myeloid-macrophage lineage. The staining for LacZ (BM tag) should be done in combination with anti-CD11b or anti-F4/80 or some other marker that indicates the macrophage or myeloid nature of BM-derived cells. Otherwise, the key conclusion is not fully supported.

3)     The authors concluded that the recruited macrophages are not proliferative but they used only one Ki-67 antibody and did not show validation of immunoreactivity of this antibody on formalin-fixed tissues (no positive control). Also, Ki-67 antibody indicates only entry to the cell cycle, some other mitotic markers should be analyzed to reach a firm conclusion.

4)     The authors make parallels to tumor-associated macrophages (TAMs) because both lactation-induced and TAMs have immunosuppressive phenotype. This is not unreasonable but perhaps a more relevant type is “wound healing” type of macrophages responsible for tissue repair and remodeling. Both TAMs and remodeling macrophages are immunosuppressive and recruited from the BM as one can verify by FISH or staining for BM markers including markers of stem/progenitor cells. The significance of authors’ findings might be extended if they consider also the remodeling function of the recruited cells instead of focusing only on immunosuppression which simply accompanies tissue repair.

5)     The other point for Discussion is regarding the statement that TAMs are created by cancer environment (lines 385-389). However, some specimens analyzed in this study are derived from non-malignant tissues. Immunosuppressive state of blood circulating monocytes during cancer, chronic inflammation and tissue regeneration is well known suggesting that this phenotype is acquired prior to entry to the tissue. Once again, one might consider that lactation macrophages might be pre-formed in the BM to help with remodeling, which may or may not related to cancer.

Minor critiques:

1)     Lines 56-57 – there are only two main sets of macrophages derived from either embryonically embedded precursors or from circulating monocytes. Those recruited by cancers are still derived from circulating monocytes, there is no other way of recruitment.

2)     Lines 354-355 – the statement requires clarification; the mentioned properties of macrophages cannot be entirely independent from the BM contribution (based on ref. 11) as multiple studies including the present study demonstrate recruitment of these cells to breast from the BM. Whatever this ref. 11 states, this should be taken in the context with other evidence.

3)     The font size in labels for Figures 2D and 5D is too small. It is also unclear what is meant by “% Positive ducts”. Positive for what?

4)     The legend for Figure 2B does not specify the antibodies used to generate 3-color image.

5)     The legend for Figure 6C does not specify the type of tissue stained and number of mice analyzed.

6)     Line 391 contains a typo (should be “what”)

Author Response

Major critiques:

  1. The bone marrow (BM) origin of mammary gland macrophages is convincing; however, the numbers of CD68+ cells with and without BM markers have not been compared. This is a missed opportunity to clarify the extent of contribution of myeloid-macrophage population from the BM and local sources.

Our aim in this study was to study specifically human and murine intraductal macrophages (foam cells). In our human studies we observed strong CD68+ positivity in virtually 100% of the intraductal foam cells. By X,Y chromosome FISH, virtually 100% of these same intraductal foam cells showed evidence of a donor bone marrow origin. Therefore in the human studies virtually 100% of the intraductal foam cells were of bone marrow donor origin and none were of local origin.

We completely agree with the suggestion of the reviewer that we can strengthen our conclusions with regard to the origin of the murine intraductal macrophages by comparing the number of CD68+ intraductal foam cells with and without BM markers. We have now conducted dual labelling of the intraductal foam cells which expressed the β-galactosidase (LacZ) tag with an alkaline phosphatase-conjugated secondary goat antibody to rabbit anti-mouse CD68 detected by 1-Step™ NBT/BCIP Substrate Solution, giving a red color.  Dual blue / red signals were observed within the same intraductal foam cells, suggesting that in the murine studies also that virtually 100% of the myeloid-macrophage lineage-derived intraductal foam cell population was of bone marrow donor origin and none were of local origin (Figure 7D, left lower inset). Although we did note CD68+ cells that were bone marrow-tag negative and CD68- cells that were bone marrow-tag positive, they were located within the mammary stroma and not within the intraductal lumens. Undoubtedly they represented respectively resident macrophages of local origin as well as non-macrophages, eg. neutrophils and lymphocytes of bone marrow origin. We did of course observe β-galactosidase (LacZ)+ positive cells within mammary stroma (Figure 7B) which could represent bone marrow-derived macrophages in transit to become intraductal foam cells. We have expanded the Materials and methods section, provided the results of these additional experiments in the Results section and modified the Figure legends section of the revised manuscript.

  1. Figure 7 shows examples of cells from BM but there is no verification that these cells (or part of them) are from the myeloid-macrophage lineage. The staining for LacZ (BM tag) should be done in combination with anti-CD11b or anti-F4/80 or some other marker that indicates the macrophage or myeloid nature of BM-derived cells. Otherwise, the key conclusion is not fully supported.

Our aim in this study was to study specifically human and murine intraductal macrophages (foam cells), defined by both their intraductal location as well as their “foamy” appearance characterized by numerous intracytoplasmic vacuoles. We do agree that verification of the intraductal foam cells that were donor bone marrow-tagged should be further confirmed as being from the myeloid-macrophage lineage and so we have now conducted dual labelling of the intraductal foam cells which expressed the β-galactosidase (LacZ) tag with an alkaline phosphatase-conjugated secondary goat antibody to rabbit anti-mouse CD11b detected by 1-Step™ NBT/BCIP Substrate Solution. Dual blue / red signals were observed within the same intraductal foam cells, suggesting that in these murine studies virtually 100% of the intraductal foam cells were of both bone marrow donor origin as well as myeloid-macrophage lineage (Figure 7D, right upper inset). We have expanded the Materials and methods section, provided the results of these additional experiments in the Results section and modified the Figure legends section of the revised manuscript.

  1. The authors concluded that the recruited macrophages are not proliferative but they used only one Ki-67 antibody and did not show validation of immunoreactivity of this antibody on formalin-fixed tissues (no positive control). Also, Ki-67 antibody indicates only entry to the cell cycle, some other mitotic markers should be analyzed to reach a firm conclusion.

We did show validation of Ki-67 immunoreactivity on formalin-fixed tissues with a positive control because in Figure 1C, we depict proliferating DCIS showing strong nuclear Ki-67 immunoreactivity surrounding central luminal intraductal foam cells (arrow) exhibiting negative Ki-67 immunoreactivity.

We completely agree, however, with the suggestion of the reviewer, that using other mitotic markers and demonstrating similar negative immunoreactivities would strengthen our conclusions that intraductal macrophages (foam cells) are not proliferative. So we investigated, in addition to Ki-67, PCNA and MCM-2, two other markers expressed at different stages of the cell cycle but never during G0 and observed that intraductal foam cells are completely negative for PCNA and MCM-2 (Figure 1C, left lower and right upper insets) suggesting that they are, in fact, in G0 and not proliferative. We have expanded the Materials and methods section, provided the results of these additional experiments in the Results section and modified the Figure legends section of the revised manuscript.

  1. The authors make parallels to tumor-associated macrophages (TAMs) because both lactation-induced and TAMs have immunosuppressive phenotype. This is not unreasonable but perhaps a more relevant type is “wound healing” type of macrophages responsible for tissue repair and remodeling. Both TAMs and remodeling macrophages are immunosuppressive and recruited from the BM as one can verify by FISH or staining for BM markers including markers of stem/progenitor cells. The significance of authors’ findings might be extended if they consider also the remodeling function of the recruited cells instead of focusing only on immunosuppression which simply accompanies tissue repair.

Our study focuses on intraductal macrophages (foam cells) which lie within lactating ducts and juxtaposed to ductal neoplastic processes. This is why we initially raised their putative role in immunosuppression and not wound healing per se. But certainly the macrophages that end up as intraductal foam cells but begin as circulating bone marrow-derived monocytes first have to enter the stromal compartment of the breast. At that stage they may, in fact, show a strikingly different phenotype including both proliferative and wound healing potential responsible for both tissue repair and remodeling. We appreciate the points raised by this reviewer and have incorporated these into the Discussion section of our revised manuscript.   

  1. The other point for Discussion is regarding the statement that TAMs are created by cancer environment (lines 385-389). However, some specimens analyzed in this study are derived from non-malignant tissues. Immunosuppressive state of blood circulating monocytes during cancer, chronic inflammation and tissue regeneration is well known suggesting that this phenotype is acquired prior to entry to the tissue. Once again, one might consider that lactation macrophages might be pre-formed in the BM to help with remodeling, which may or may not related to cancer.

The points raised by the reviewer are provocative and have now been raised in the Discussion section of the revised manuscript. As has been stated, our study focuses on the intraductal foam cell and the evidence supporting its exclusive myeloid-macrophage lineage and bone marrow origin. Since the phagocytic appearance of the foam cell is not apparent until it enters its intraductal location, other phenotypic properties of this cell, eg immunosuppression, may only be expressed as well only when it reaches its intraductal location. Alternatively, the subset of bone marrow-derived myeloid-macrophage lineage cells destined to become foam cells which also participate in either lactation or as TAMs in the setting of DCIS may have acquired its phenotypic properties at an earlier stage ranging from its bone marrow origin, circulating monocyte pool or entrance into the mammary stroma. All these possibilities merit investigation in future studies.

Minor critiques:

  1. Lines 56-57 – there are only two main sets of macrophages derived from either embryonically embedded precursors or from circulating monocytes. Those recruited by cancers are still derived from circulating monocytes, there is no other way of recruitment.

We have rewritten this portion in our Introduction section of our revised manuscript to properly credit that mammary macrophages, especially those associated with lactation and mammary cancers, reflect the bone marrow contribution.

  1. Lines 354-355 – the statement requires clarification; the mentioned properties of macrophages cannot be entirely independent from the BM contribution (based on ref. 11) as multiple studies including the present demonstrate recruitment of these cells to breast from BM. Whatever ref. 11 states, this should be taken in the context with other evidence.

We have rewritten this portion of the Discussion in our revised manuscript to properly emphasize that mammary macrophages especially those associated with lactation and mammary carcinomas reflect the bone marrow contribution.

  1. The font size in labels for Figures 2D and 5D is too small. It is also unclear what is meant by “% Positive ducts”. Positive for what?

We have enlarged the font size for both figures. % ducts and % positive ducts are the overall percentage of ducts that contain intraductal macrophages (foam cells). Foam cells and # of foam cells/duct are the average numbers of intraductal macrophages (foam cells)/duct in the ducts containing foam cells. We have added this clarification to both figure legends (Figure 2D and Figure 5D) in the Figure legends section of our revised manuscript.

  1. The Figure 2B legend does not specify the antibodies used to generate 3-color image.

We have revised the figure legend to specify the antibodies used to generate the 3-color image that reflect the following.  Our specific TMA algorithms which carried out virtual alignment, image processing, and the application of the epithelial recognition algorithms (ERAs) and specific recognition algorithms (SRAs) which recognized ductal profiles based initially on circumferential maspin myoepithelial immunoreactivity (A), subsequent imaging processing defined ductal profiles based on circumferential epithelial E-cadherin (green fluorescence), juxtaposed myoepithelial maspin (red fluorescence) and DAPI nuclear (blue fluorescence) (B) and recognition of intraductal macrophage CD68+ immunoreactivity (C) were able to calculate frequency and density of foam cells (D). Scale bars are provided. We have also expanded the Materials and methods section and provided these additional experiments in the Results section of the revised manuscript.

  1. The legend for Figure 6C does not specify the tissue type stained and # mice analyzed.

We have revised the figure legend to reflect the following. Bone marrow of ROSA 26 donor mice, harvested by femoral flushing (A) and marked with the β-galactosidase (LacZ) reporter (B) was transplanted into recipient mice and successful engraftment marked by at least 50% engraftment when sections of murine bone marrow were analyzed 60 days after bone marrow transplant (C). Gross photographs and photomicrographs are depicted from single representative cases but 8/10 mice showed successful bone marrow engraftment. Scale bars are provided. We have revised this legend in the Figure legends section of our revised manuscript.

  1. Line 391 contains a typo (should be “what”).

This “typo” has been corrected in the Discussion section of our revised manuscript.

Round 2

Reviewer 2 Report

Comments and Suggestions for Authors

The revised version addressed my concerns.